# LEARNING GLOBAL SPATIAL INFORMATION FOR MULTI-VIEW OBJECT-CENTRIC MODELS

## ABSTRACT

Recently, several studies have been working on multi-view object-centric models, which predict unobserved views of a scene and infer object-centric representations from several observation views. In general, multi-object scenes can be uniquely determined if both the properties of individual objects and the spatial arrangement of objects are specified; however, existing multi-view object-centric models only infer object-level representations and lack spatial information. This insufficient modeling can degrade novel-view synthesis quality and make it difficult to generate novel scenes. We can model both spatial information and object representations by introducing hierarchical probabilistic model, which contains a global latent variable on top of object-level latent variables. However, how to execute inference and training with that hierarchical multi-view object-centric model is unclear. Therefore, we introduce several crucial components which help inference and training with the proposed model. We show that the proposed method achieves good inference quality and can also generate novel scenes.

## 1 INTRODUCTION

Extracting representations of individual objects from direct perception (e.g., visual input) is called object-centric representation learning. It is considered to be effective in many aspects such as robotics (Devin et al., 2018; Veerapaneni et al., 2019; Kulkarni et al., 2019), reasoning (Ding et al., 2021), and sample efficiency (Watters et al., 2019a). Among them, the deep generative model (DGM)-based method is one of the promising directions, as it can infer rich information about objects in a fully unsupervised manner, and some of its models can also generate novel scenes (Greff et al., 2017; Locatello et al., 2020; Burgess et al., 2019; Greff et al., 2019; Engelcke et al., 2020; Eslami et al., 2016; Crawford & Pineau, 2019; Jiang & Ahn, 2020).

Recently, several studies have been working on multi-view object-centric models, which predict unobserved views of a scene: novel view synthesis, and infer object-centric representations from several observation views (Nanbo et al., 2020; Chen et al., 2021). This can be also regarded as an object-centric expansion of the Generative Query Network (GQN) (Eslami et al., 2018). In order to represent multi-object scenes, the properties of individual objects and the spatial arrangement of objects should be specified. However, existing multi-view object-centric methods only explicitly model representations of individual objects, and each of them includes their own spatial information separately. Namely, relationship between objects is not represented in this case. This modeling can degrade novel view synthesis and segmentation, because the model need to solve occlusions and spatial ambiguity virtually without the prior knowledge about objects' spatial relationship. This can be serious especially when the number of available observation views is limited. In addition, this modeling can also lead to inability to generate physically plausible novel scenes, because they need to place objects independently, which often brings about collisions and misplacement of objects.

Spatial information of the whole scene can be represented by introducing another latent variable besides object-level latent variables. This variable can be independent of object-level latent variables, or can be modeled as a global latent on top of object-level latent variables. In the former formulation, those latent variables need to be disentangled during training, which can often be challenging. From another point of view, it is common assumption in cognitive science that humans recognize the whole and parts, or objects, separately (Schmidt, 2009). In feature integration theory (Treisman & Gelade, 1980), visual feature maps become "master map of locations" combined with positions of

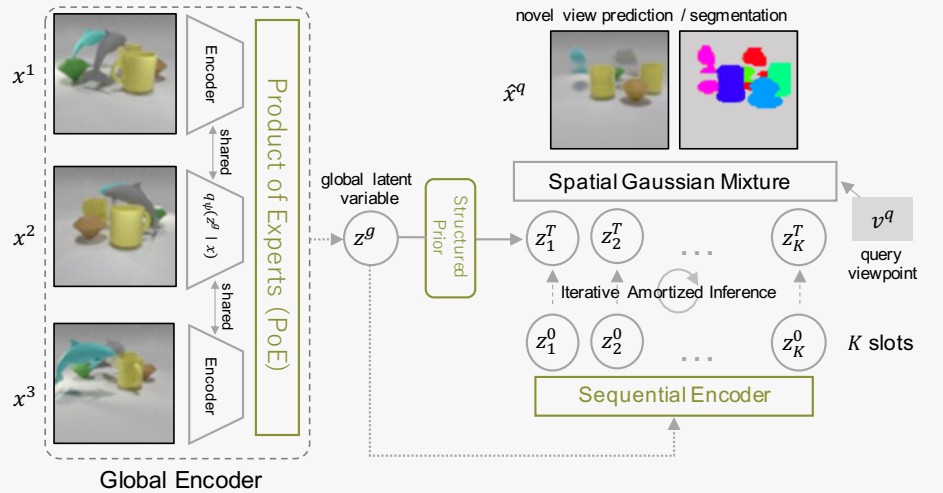

Figure 1: Overview of the proposed model. In this example, the model predicted an image and corresponding segmentation masks from unobserved query viewpoint $\mathbf{v}^q$, using three observation views $\mathbf{x}^1$-$\mathbf{x}^3$. $\mathbf{z}^g$ is a global latent variable that represents spatial relationship between objects, and $\mathbf{z}_k$ is object-level latent variables that represent each object. Note that the $\hat{x}^q$ and segmentation shown in this figure are the actual results from our model.

objects in visual space. Then, attention binds those features to recognize individual objects. Modeling the spatial arrangement of objects as a global latent variable closely adheres to this point of view due to its hierarchical property. For the above reasons, we believe that introducing a global latent variable is more natural step for improving object-centric generative models.

In this paper, we build a new multi-view object-centric model which has a latent variable for global spatial information to improve inference: novel view synthesis and segmentation. In addition, our model can also generate physically plausible novel scenes. The problem is that how to execute inference and training with the proposed model is unclear. In regards to this point, we propose several crucial components: Global Encoder, Sequential Encoder and Structured Prior. Firstly, the model needs to infer global representation from variable number of input views, thus we introduced an encoder using Product of Experts (PoE) (Hinton, 2002), which we refer to as Global Encoder. Secondly, the structured, complex posterior of object-level latent variables need to be sampled including spatial relationship between objects in order to generated physically plausible novel scenes. Therefore, we introduced a learnable prior called Structured Prior, which is implemented by Transformer (Vaswani et al., 2017). Lastly, we introduced an encoder using auto-regressive model called Sequential Encoder. This stabilize inference and training with the hierarchical probabilistic model of the proposed method.

The proposed model has representation about the whole, or spatial information of a scene and has representations about a part of a scene (i.e., individual properties of objects); hence, we call this model the Whole-Part Representation Learning Model for Object-Centric 3D Scene Inference and Sampling (**WeLIS**).

**Contributions:**

- We introduce a multi-view object-centric model with a global latent variable which represents the spatial configuration of objects in a scene.
- We introduce essential components for introducing a global latent variable as well as stable learning and inference for our multi-view object-centric model.
- We show that the proposed model performs well in terms of inference quality, and can generate novel scenes and corresponding segmentation masks properly, unlike existing methods.

Table 1: Comparison of multi-object-multi-view (MOMV) related methods.

| Model | Object-Centric | Novel View Synthesis | Inference | Sampling (Novel Scene Generation) |
|---|---|---|---|---|
| GQN | ✗ | ✓ | ✓ | ✓ (Kosiorek et al., 2021) |
| ObSuRF, uORF [1] | ✓ | ✓ | ✓ | ✗ |
| ROOTS | ✓ | ✓ | ✓ | ✗ |
| MulMON | ✓ | ✓ | ✓ | ✗ [2] |
| WeLIS (Ours) | ✓ | ✓ | ✓ | ✓ |

[1]These methods are not generative models.
[2]MulMON can generate objects independently, but this leads to an ambiguous, blurred image.

## 2 RELATED WORK

**Object-Centric Models**:   The objective of object-centric representation learning is to obtain representations of each object from visual input. It is applicable to many domains such as robotics (Devin et al., 2018; Florence et al., 2019; Veerapaneni et al., 2019), reasoning (Ding et al., 2021) and reinforcement learning (Watters et al., 2019a). One of the promising directions of object-centric representation learning is variational autoencoder (VAE)-based (Kingma & Welling, 2013) methods, as they can obtain rich information about objects that can also be disentangled, and some of them can generate novel scenes outside of the empirical distribution (Burgess et al., 2019; Greff et al., 2019; Engelcke et al., 2020; Eslami et al., 2016; Crawford & Pineau, 2019; Jiang & Ahn, 2020). VAE-based object-centric models are often referred to as "scene interpretation models" due to their inference ability. Some models are capable of object-centric generation (Dai et al., 2019; Nguyen-Phuoc et al., 2020; Niemeyer & Geiger, 2021), however, these methods focus on generation and not for inference.

**Multi-View Object-Centric Models**:   Lately, several studies have been working on multi-view scene interpretation models (Nanbo et al., 2020; Chen et al., 2021). These methods predict images of a 3D scene from arbitrary viewpoints in an object-centric way, using several observation views as input. This problem setting can be regarded as an object-centric expansion of GQN family (Eslami et al., 2018; Tobin et al., 2019; Kosiorek et al., 2021). The above mentioned single-view setting is referred to as multi-object-single-view (MOSV), and the multi-view setting is defined as multi-object-multi-view (MOMV) in (Nanbo et al., 2020).

Table 1 compares methods related to the MOMV setting. Novel view synthesis stands for prediction of unobserved viewpoints, and novel scene generation stands for random generation of novel scenes. Among these models, ROOTS and MulMON (Nanbo et al., 2020; Chen et al., 2021) are the only generative models that work on a MOMV setting. In addition to this, O3V (Henderson & Lampert, 2020) have a similar problem setting, but it takes fixed number of viewpoints as input. ObSuRF (Stelzner et al., 2021) and uORF(Yu et al., 2021) also work on MOMV setting, but these are not generative models, and are based on Slot Attention (Locatello et al., 2020) and NeRF (Mildenhall et al., 2020). Therefore these methods can not generate novel samples and does not obtain low dimensional representation space, or latent space.

In this paper, we introduce a new MOMV model with a global latent variable, which can potentially represents the spatial arrangement of objects in a scene. The proposed model is the only MOMV model that can generate novel scenes. In this paper, we set MulMON as our baseline model due to the following reasons. Firstly, MulMON uses segmentation masks to represent objects' regions unlike ROOTS which uses bounding boxes. Bounding box-based methods, which originate from (Eslami et al., 2016) tend to be more sensitive to object size and shape (Engelcke et al., 2021). Secondly, MulMON is more robust in terms of the number of observation views required. Although we built our model based on MulMON, a similar approach can be applied to ROOTS and other methods.

Introducing the global latent variable itself has been done in some studies in various areas (Vasco et al., 2020; Akuzawa et al., 2021). In particular, GNM (Jiang & Ahn, 2020) introduces a global latent variable to bounding box-based single-view object-centric model for random generation. However, how to execute stable inference and training with the proposed MOMV model is unclear, and is thus the points of investigation in this paper.

## 3 METHOD

Our objective was to introduce a MOMV model that has both object-level representations and the spatial arrangement of the scene to obtain further inference quality and novel scene generation ability.

We achieved this by introducing a global latent variable for global spatial information on top of object latent variables, which is formulated as hierarchical probabilistic model. In this paper, we introduce several crucial components to realize stable inference and training with this MOMV model. An overview of the proposed model is shown in Figure 1.

In this section, we first explain the problem setting in this paper, and then the probabilistic model of our proposed model. Lastly, the training method and additional details of the introduced components are explained.

### 3.1 PROBLEM SETTING

The target of this research is 3D scenes with multiple objects, and thus a dataset with $N$ data points can be defined as $\mathcal{D} = \{\mathbf{X}_1, \cdots, \mathbf{X}_N\}$, where $\mathbf{X}_i = \{(\mathbf{x}_i^1, \mathbf{v}_i^1), ..., (\mathbf{x}_i^M, \mathbf{v}_i^M)\}$, which corresponds to one scene with $M$ views, and $\mathbf{v}$ is a viewpoint vector with arbitrary dimensions, which are mostly three.

The task is to predict queried unobserved views of the scene given several observations $\{(\mathbf{x}_i^1, \mathbf{v}_i^1), \cdots, (\mathbf{x}_i^{N_{obs}}, \mathbf{v}_i^{N_{obs}})\}$, and infer $K$ representations each of which correspond to an object in the scene $\mathbf{Z} = \{\mathbf{z}_1, \cdots, \mathbf{z}_K\}$. In practice, the $M$ views in $\mathbf{X}_i$ is split into $N_{obs}$ observations and $N_{qry}$ queries in the training process. $N_{obs}$ can be varied in every iteration for generalization.

### 3.2 PROBABILISTIC MODEL

**Generative Model**: As mentioned above, we introduced the hierarchical probabilistic model with global latent $\mathbf{z}^g$ on top of object-level latent variables $\mathbf{z} = \{\mathbf{z}_1, \cdots, \mathbf{z}_K\}$. In the following sections, object-level latent variables are abbreviated as $\mathbf{z}$, as required. The marginal likelihood is thus:

$$p_\gamma(\mathbf{x} \mid \mathbf{v}) = \int \int p_\theta(\mathbf{x}|\mathbf{z}, \mathbf{v})\, p_\phi(\mathbf{z} \mid \mathbf{z}^g)\, p(\mathbf{z}^g)\, d\mathbf{z}d\mathbf{z}^g, \tag{1}$$

where $\gamma = \{\theta, \phi\}$. The generative model $p_\theta(\mathbf{x}|\mathbf{z}, \mathbf{v})$ is a spatial Gaussian mixture model derived from prior works (Nanbo et al., 2020; Burgess et al., 2019) that can be written as:

$$p_\theta(\mathbf{x}|\mathbf{z}, \mathbf{v}) = \prod_{i=1}^{D} \sum_{k=1}^{K} p_\theta(C_i = k \mid \mathbf{z}_k, \mathbf{v}) \otimes p_\theta(x_{ik}|\mathbf{z}_k, \mathbf{v}), \tag{2}$$

where $D \in \mathbb{N}$ is the number of image pixels, and $K$ is the number of mixture components which are often referred to as "slots". The decoder has parameters $\theta$ and implemented by a broadcast decoder (Watters et al., 2019b). $\mathbf{z}_k$ and $\mathbf{v}$ are first concatenated and projected onto a single vector $\mathbf{f}_k$ by MLP and then fed into the decoder. The decoder generates $K$ components $\{\mathbf{x}_1, \cdots, \mathbf{x}_K\}$ and corresponding segmentation masks from $\{\mathbf{f}_1, \cdots, \mathbf{f}_K\}$.

The second term in the r.h.s of Equation 1 $p_\phi(\mathbf{z} \mid \mathbf{z}^g)$ is a learnable prior which we refer to as the Structured Prior, and the third term is a standard Gaussian distribution. The Structured Prior is explained in Section 3.3.

**Inference Model**: We used amortized variational inference (Kingma & Welling, 2013) and computed variational approximate posteriors because the marginal likelihood in Equation 1 is intractable. We factorized the variational posteriors as:

$$q_\psi(\mathbf{z}, \mathbf{z}^g \mid \mathbf{X}) = q_\psi(\mathbf{z}_1, \cdots, \mathbf{z}_K \mid \mathbf{z}^g)\, q_\psi(\mathbf{z}^g \mid \mathbf{X}), \tag{3}$$

where $\mathbf{X} = \{(\mathbf{x}_1, \mathbf{v}_1), \ldots, (\mathbf{x}_{N_{obs}}, \mathbf{v}_{N_{obs}})\}$. The first RHS term and second RHS term are approximate posteriors of $\mathbf{z}$ and $\mathbf{z}^g$ respectively. We refer to these encoders as Sequential Encoder and Global Encoder.

The Global Encoder adopts PoE (Hinton, 2002) to deal with a variable number of input observation views $N_{obs}$. In particular, we used PoE with prior expert (Wu & Goodman, 2018). In addition to this, we applied normalizing flow (NF) (Rezende & Mohamed, 2015) to this inference model. Although this is optional, the posterior should be expressive in order to represent spatial configuration of complex 3D multi-object scenes. Therefore, the variational approximate posterior is:

$$\mathbf{z}^g \sim q_\psi\left(\mathbf{z}^g \mid \mathbf{X}\right) = \left(p(\mathbf{z}_1^g) \prod_{v=1}^{N_{obs}} q_\psi(\mathbf{z}_1^g \mid \mathbf{x}_v, \mathbf{v}_v)\right) \prod_{t=1}^{T} \left|\det \frac{\partial f_\psi}{\partial \mathbf{z}_t^g}\right|^{-1}, \tag{4}$$

where $T$ is the number of transformations of NF, and $N_{obs}$ is the number of available observation views. We used planar flow for its simplicity, but any methods can be applied and can improve the results potentially. Thus, the transform $f_\psi$ in the above equation is

$$f_\psi(\mathbf{z}_{i+1}) = \mathbf{z}_i + \mathbf{u}_i \cdot h(\mathbf{w}^T \mathbf{z}_i + \mathbf{b}_i).$$

The Sequential Encoder was implemented in auto-regressive manner using LSTM (Hochreiter & Schmidhuber, 1997). We tried several architectures, but non-auto-regressive models were unstable as long as we have tested. Firstly, this auto-regressive part infers $K$ latent variables $\mathbf{z}_1^0, \ldots, \mathbf{z}_K^0$ and then, these are updated using iterative amortized inference (IAI) (Marino et al., 2018) as MulMON and IODINE (Greff et al., 2019) do. However, we omitted image-sized inputs as in (Emami et al., 2021). Thus, the Sequential Encoder can be defined as:

$$\mathbf{z}_1^0, \ldots, \mathbf{z}_K^0 \sim \prod_{k=1}^{K} q_\psi(\mathbf{z}_k^0 \mid \mathbf{z}_{1:k}^0, \mathbf{z}^g), \tag{5}$$

$$\mathbf{z}_k^{i+1} = \mathbf{z}_k^i + f_\psi(\mathbf{z}_k^i, \mathbf{z}^g, \nabla_{\mathbf{z}_k^i} \mathcal{L}_{IAI}^i), \tag{6}$$

where $\nabla_{\mathbf{z}_k^i} \mathcal{L}_{IAI}^i$ is a gradient of negative log-likelihood in that iteration, and $f_\psi$ is a refinement network that is implemented by MLP followed by LSTM. The posterior can be updated arbitrary times, and we set $L = 5$ through this paper. The details of $\mathcal{L}_{IAI}^i$ will be explained in the next section.

**Training**: All the parameters in this model can be trained end-to-end by maximizing evidence lower bound (ELBO) on the log-marginal likelihood $\log p(\mathbf{x}_1, \cdots, \mathbf{x}_N)$. The ELBO of the proposed model is

$$\mathcal{L}_{ELBO} = E_{q(\mathbf{z},\mathbf{z}^g|\mathbf{X})} \left[\log \frac{p_\theta(\mathbf{x}|\mathbf{z},\mathbf{v})p_\phi(\mathbf{z} \mid \mathbf{z}^g)p(\mathbf{z}^g)}{q_\psi(\mathbf{z} \mid \mathbf{z}^g)q_\psi(\mathbf{z}^g \mid \mathbf{X})}\right] \tag{7}$$

$$= E_{q(\mathbf{z},\mathbf{z}^g|\mathbf{X})} [\log p_\theta(\mathbf{x} \mid \mathbf{z},\mathbf{v})]$$
$$- KL\left[q_\psi(\mathbf{z}^g \mid \mathbf{X}) \mid\mid p(\mathbf{z}^g)\right] - E_{q_\psi(\mathbf{z}^g|\mathbf{X})}\left[KL\left[q_\psi(\mathbf{z} \mid \mathbf{z}^g) \mid\mid p_\phi(\mathbf{z} \mid \mathbf{z}^g)\right]\right].$$

Here, KL stands for Kullback–Leibler divergence. Then, introducing two coefficient $\beta_1$ and $\beta_2$ for balancing KL terms, the objective can be written as

$$\mathcal{L} = E_{q(\mathbf{z},\mathbf{z}^g|\mathbf{X})} [\log p_\theta(\mathbf{x} \mid \mathbf{z},\mathbf{v})]$$
$$- \beta_1 KL\left[q_\psi(\mathbf{z}^g \mid \mathbf{X}) \mid\mid p(\mathbf{z}^g)\right] - \beta_2 E_{q_\psi(\mathbf{z}^g|\mathbf{X})}\left[KL\left[q_\psi(\mathbf{z} \mid \mathbf{z}^g) \mid\mid p_\phi(\mathbf{z} \mid \mathbf{z}^g)\right]\right]. \tag{8}$$

Moreover, due to the iterative process of IAI, negative log-likelihood is computed as

$$E_{q(\mathbf{z},\mathbf{z}^g|\mathbf{X})} [\log p_\theta(\mathbf{x} \mid \mathbf{z},\mathbf{v})] = \frac{2}{L(L+1)} \sum_{i=0}^{L-1} (i+1)\mathcal{L}_{IAI}^i, \tag{9}$$

$$\mathcal{L}_{IAI}^i = \mathbb{E}_{\mathbf{z}^i \sim q(\mathbf{z}^i, \mathbf{z}^g|\mathbf{X})} \left[\log p_\theta(\mathbf{x}|\mathbf{z}^i, \mathbf{v})\right], \tag{10}$$

where, $L$ is the number of iterations in IAI. As shown in Equation 9, log-likelihoods in the later steps of iterative refinement are emphasized as in MulMON and IODINE.

In addition to this, we used GECO (Rezende & Viola, 2018) as in some existing methods (Engelcke et al., 2020; Emami et al., 2021). GECO reinterprets the maximization of ELBO as a constrained

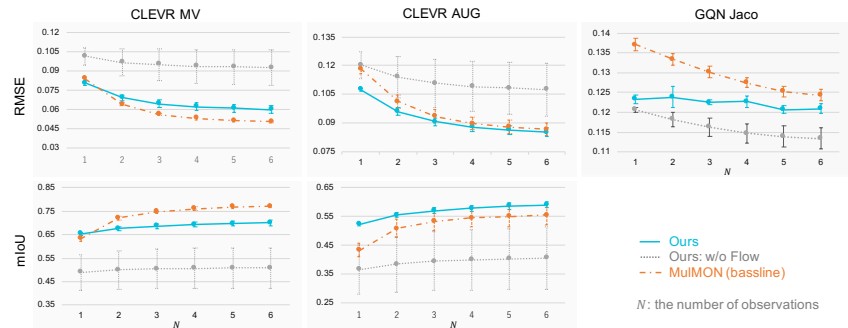

Figure 2: Quantitative evaluation of novel view synthesis. Lower is better for root mean squared error(RMSE), and higher is better for mean intersection over union (mIoU). There is no mIoU for GQN Jaco dataset because it does not provide ground truth segmentation masks.

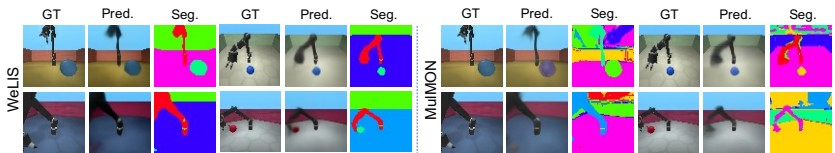

Figure 3: Novel view synthesis results on GQN Jaco with four difference scenes. These images are prediction of unobserved query views. The number of observation views is three ($N = 3$).

optimization problem that takes KL terms as the main objective and log-likelihood as the constraint. This optimization is done by the method of Lagrange multipliers, namely, the coefficient of KL terms $\beta$ is regarded as a Lagrange multiplier, and automatically tweaked during training. GECO is not necessary for our model, but proper balancing of $\beta$ further improves random generation quality.

The reinterpreted optimization objective thus becomes

$$\underset{\psi,\phi}{\arg\min} \, KL\left[q_\psi(\mathbf{z}^g \mid \mathbf{x}_{obs}, \mathbf{v}_{obs}) \mid\mid p(\mathbf{z}^g)\right] + E_{q_\psi(\mathbf{z}^g|\mathbf{X})}\left[KL\left[q_\psi(\mathbf{z} \mid \mathbf{z}^g) \mid\mid p_\phi(\mathbf{z} \mid \mathbf{z}^g)\right]\right] \quad (11)$$

$$\text{such that} \quad -E_{q_\psi(\mathbf{z},\mathbf{z}^g|\mathbf{X})}\left[\log p_\theta(\mathbf{x} \mid \mathbf{z}, \mathbf{v})\right] \leqq R, \quad (12)$$

where $R \in \mathbb{R}$ is the maximum allowed negative log-likelihood, which is a hyperparameter in GECO instead of $\beta$. Although all the parameters can be trained end-to-end, we found that end-to-end training including the parameters of Structured Prior $\phi$ is slightly unstable and often degrades novel scene generation. Thus, we introduced a separate training strategy, in which $\phi$ is updated after the training of the main model. Detail of this method is explained in the next section.

### 3.3 STRUCTURED PRIOR

In this section, we explain the detail of the Structured Prior and its separate training strategy.

First of all, the Structured Prior is defined as $p_\phi(\mathbf{z}_1^L, \cdots, \mathbf{z}_K^L \mid \mathbf{z}^g)$. As we mentioned in Equation 5, the latent variables of each slots are iteratively updated by IAI in the inference process. However, it makes sampling from the updated posterior $\mathbf{z}_1^L, \cdots, \mathbf{z}_K^L$ difficult. Though IAI updates latent variables using reconstruction error and auxiliary inputs, generative model needs to blindly capture this process without using them.

Therefore, in order to model this process, we implemented Structured Prior as Transformer (Vaswani et al., 2017). Firstly, initial $K$ variables $\mathbf{z}_k^0$ are computed from $\mathbf{z}^g$ using the Sequential Encoder $q_\psi(\mathbf{z}^0 \mid \mathbf{z}^g)$. Then, Transformer takes the $K$ latent variables as input, and approximates updated ones $\mathbf{z}_k^L$. Note that the parameter of Sequential Encoder is shared with inference.

**Separate Training**: As we mentioned above, we introduced a separate training method for the Structured Prior. In a first stage, WeLIS is trained without the parameters of Structured Prior $\phi$.

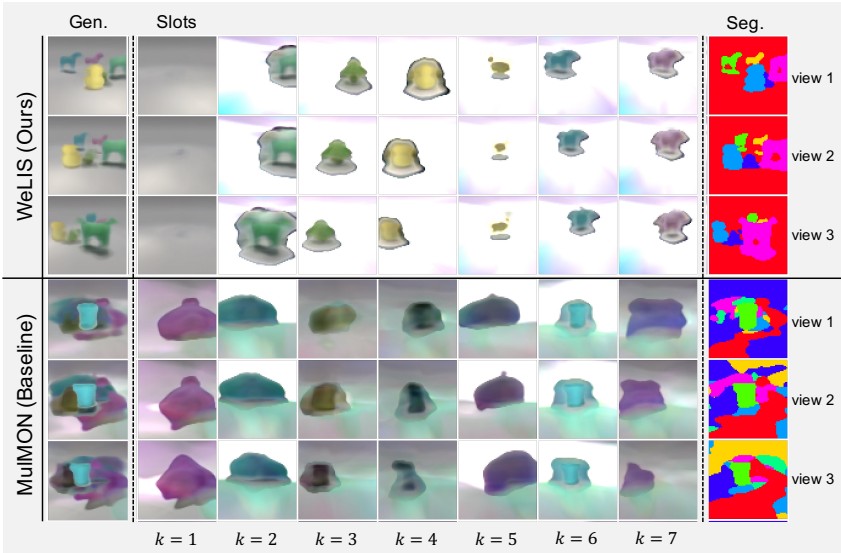

Figure 4: Component-by-component random generation results of WeLIS and the baseline. Each row corresponds to a different query viewpoint. The first column shows generated images, and the next seven columns show each slot ($\mathbf{x}_k$, $k \in \{1, \cdots, K\}$), while last column shows generated segmentation masks.

Namely, $\phi$ is frozen and $\beta_2$ in Equation 7 is set to zero or the second term of Equation 11 is excluded if we use GECO. Then, as a second stage, $\phi$ is updated while other parameters are frozen.

## 4 EXPERIMENTS

We evaluated our model with three different datasets derived from the MulMON paper (Nanbo et al., 2020). CLEVR Multi-View (CLEVR MV) is a multi-view version of the CLEVR dataset (Johnson et al., 2017), and CLEVR Augmented (CLEVR AUG) extends CLEVR MV with more complex and various objects for additional difficulty. GQN Jaco is a simulation based robot arm dataset introduced originally in (Eslami et al., 2018). All the datasets are used in $64 \times 64$ resolution [1], and we trained all the models with Adam optimizer for $300k$ iterations. We used official implementation of MulMON for reproduction. Up to six views are given as observation during training.

First, we evaluated inference quality quantitatively and show that WeLIS outperforms MulMON in many cases. Then, we show WeLIS can also generate novel scenes, in contrast to baseline models. Lastly, we show ablation results and also investigate how global latent variable $\mathbf{z}^g$ works. In addition, additional ablation studies, samples, and the results of downstream task are shown in Appendix B.

### 4.1 NOVEL VIEW SYNTHESIS AND SEGMENTATION

In this section, we evaluate novel view synthesis and segmentation quality. Figure 2 shows quantitative evaluation of three models: WeLIS, WeLIS without NF (ablation) and MulMON. Novel view synthesis is evaluated by Root Mean Squared Error (RMSE), and segmentation is evaluated by mean Intersection over Union(mIoU). Random four seeds are used to calculate these scores. As GQN Jaco does not provide ground truth segmentation, we instead show novel view synthesis results in Figure 3.

We can see that WeLIS outperforms MulMON in CLEVR AUG, GQN Jaco and CLEVR MV with a small number of observation views $N$. WeLIS is relatively stable in terms of $N$, which means that WeLIS is good at inferring whole scene from limited, partial observation. Introducing global

---

[1]According to the supplemental material, MulMON uses CLEVR AUG in $128 \times 128$ resolution, but we used in $64 \times 64$ for consistent comparison.

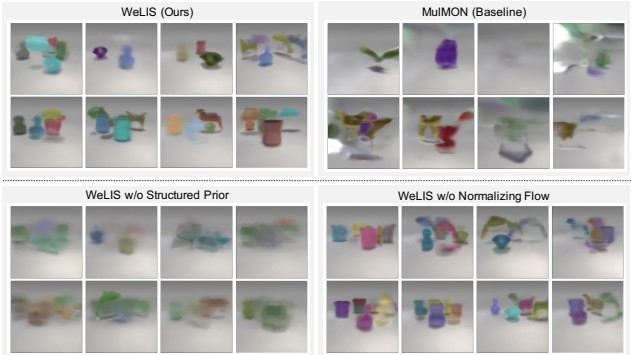

Figure 5: Random generation results from WeLIS and MulMON(baseline). The top row shows results from WeLIS and MulMON. The bottom row shows WeLIS without Structured Prior and without normalizing flow for ablation study. We generated these images by randomly sampling eight $\mathbf{z}_g$ for WeLIS, and by sampling each slot independently for MulMON. Note that the same set of $\mathbf{z}_g$ is used in the full model(top left) and in the ablation of the Structured Prior(bottom left).

Table 2: Evaluation of FID score on three models: WeLIS(Ours), WeLIS without Normalizing Flow and MulMON. Lower is better.

| Datasets | WeLIS (Ours) | WeLIS w/o NF | MulMON |
|---|---|---|---|
| CLEVR MV | 125.8 ±1.7 | **117.2** ±5.9 | 157.9 ±8.4 |
| CLEVR AUG | **83.3** ±1.7 | 134.2 ±25.8 [1] | 168.6 ±5.0 |
| GQN Jaco | **251.7** ±2.1 | 262.1 ±11.0 | 273.7 ±10.4 |

[1]If we exclude the case where training did not converge, the score becomes $90.0 \pm 1.3$.

latent in addition to object-level latent variables leads to more precise modeling of the world, thus we consider that it helps inference with small number of observations. As for GQN Jaco (Figure 3), WeLIS captures the balls consistently with clean segmentation.

We also evaluated WeLIS without NF as ablation study. We found that NF is important for stable inference and training. Without it, the model often fall into local minima that does not properly represent one object per slot, as you can see from the relatively large standard error in Figure 2. If the training successfully finished, the model achieved decent performance close to full model.

## 4.2 NOVEL SCENE GENERATION

In this section, we evaluate novel scene generation results. As shown in Figure 4, WeLIS successfully generates objects and segmentation masks, and as a result, generates physically plausible novel scenes. On the other hand, MulMON generates blurred and spatially ambiguous images. Top row of Figure 5 shows more samples.

### 4.2.1 ABLATION STUDIES

Bottom row of Figure 5 shows ablation studies of Structured Prior and NF. Firstly, we look into Structured Prior. In this experiment, the results from full model (top left) and ablation study of Structured Prior (bottom left) are generated from same $\mathbf{z}_g$. Generated images without Structured Prior are severely blurred, however, we can see that the spatial arrangement of objects itself is similar to full model. This indicates that Structured Prior only contributes to object-level generation quality.

Secondly, bottom right images in Figure 5 shows ablation result of NF. As you can see, the quality is almost same with full model even without NF, which indicates that NF is not essential for generation.

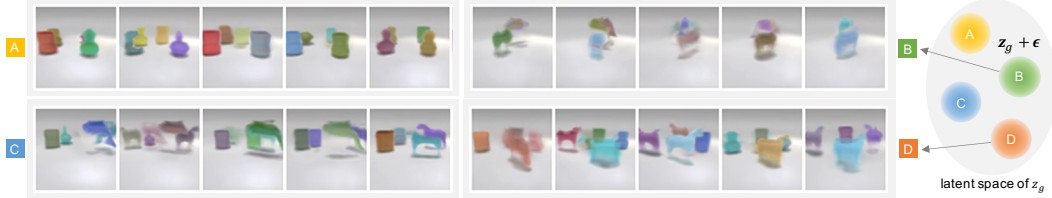

Figure 6: Random generation results from four different $\mathbf{z}_g$ (A–D) and four neighborhoods $\mathbf{z}_g + \epsilon$, where $\epsilon$ is a Gaussian noise as perturbation. Each row corresponds to a different $\mathbf{z}_g$. The left most columns of each group show the original $\mathbf{z}_g$, and other columns show different perturbations $\epsilon$. Therefore, each row represents a "cluster" of each $\mathbf{z}_g$. We can see that images in the same row share a similar spatial structure. The schematic of the latent space on the right side is just for illustration purposes.

However, as we mentioned in Section 4.1, NF is important for stable inference and training, thus it is also necessary for generation in practice.

### 4.2.2 QUANTITATIVE EVALUATIONS

We also quantitatively evaluated random generation by Fréchet Inception Distance(FID) score in Table 2. WeLIS outperforms MulMON in every dataset, and as shown in above section, WeLIS without NF achieved similar quality to full model, as long as the training succeeded. Scores without NF in CLEVR AUG and GQN Jaco are degraded because the training fails more frequently as datasets become difficult.

Poor generation quality of MulMON comes from mainly two reasons: insufficient modeling of spatial arrangement and difficulty in sampling actual posterior. In WeLIS, former one is dealt with by introducing global latent $\mathbf{z}_g$ which represents spatial information of a whole scene, and latter one is dealt with by a Structured Prior which estimates complex posterior inferred from IAI. Effectiveness of introducing $\mathbf{z}_g$ is evaluated in Section 4.3 and of Structured Prior is already investigated in Figure 5.

### 4.3 INSPECTION OF GLOBAL LATENT SPACE

To look into the representation obtained by global latent $\mathbf{z}_g$, we show random generation results from four different $\mathbf{z}_g$ (A-D) and four neighborhoods $\mathbf{z}_g + \epsilon$ of them in Figure 6. Here, $\epsilon$ is a small perturbation randomly sampled from standard normal distribution. First column in each group shows images generated from four different $\mathbf{z}_g$, and other four columns are generated with different perturbation. Thus, each four group represents a "cluster" around $\mathbf{z}_g$.

We can see that generated scenes from same $\mathbf{z}_g$ have similar spatial structure, while their objects vary. This indicates that the latent space of $\mathbf{z}_g$ obtains spatial configuration of a scene as we expected.

## 5 CONCLUSION

We introduced new multi-view object-centric model: WeLIS, which is able to perform multi-view object-centric inference and sampling in 3D scenes. We introduced several components to WeLIS, in order to adapt a global latent variable and to enable inference and training with that model. The global latent variable improved inference quality especially when the observations are limited (Figure 2) and also enabled the model to generate physically plausible novel scenes (Figure 5). We also conducted some ablation studies and investigated the representation obtained by global latent space (Figure 6).

One of the future directions of this research is to sophisticate inference methods: normalizing flow and IAI. There is a room for improvement with optimal choice of normalizing flow. In addition, we used simplified IAI without image-sized input which can reduce computational cost, however there is a possibility that this sacrificed the ability to distinguish similar, ambiguous objects such as rounded cubes and spheres in CLEVR MV dataset.

## 6   REPRODUCIBILITY STATEMENT

**Architecture**:
Detail of the network architecture is provided in Appendix A. Public version of the source code will be released soon.

**Optimization**:
We derived the training scheme from MulMON in which all models are trained for 300k gradient steps with Adam optimizer. 300k gradient steps is composed of 1000 epochs × 300 scenes. The 300 scenes are randomly picked in each iteration. Learning rate decay in GQN and MulMON is also applied to our model.

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

# A  NETWORK ARCHITECTURE

## A.1  GLOBAL ENCODER

| Type | Size/Channels | Stride | Activation | Type | Size | Activation |
|------|---------------|--------|------------|------|------|------------|
| Input | 3 | | | Input | $\mathbf{v}$ | |
| Conv 3×3 | 32 | | | Linear | 128 | ReLU |
| Conv 3×3 | 32 | | | Linear | embed_dim | None |
| Conv 3×3 | 64 | 2 | ReLU | | | |
| Conv 3×3 | 64 | | | | | |
| Linear | 256 | - | Swish | concatenation of above outputs | | |
| Linear | $\mathbf{z}^g$ | | None | reparameterization | | |
| Product of Experts | | | | | | |

Table 3: Architecture of Global Encoder. This table contains convolution network (top left), viewpoint embedding (top right), feedforward network.

Global Encoder is composed of CNN, viewpoint embedding, feedforward network for reparameterization and PoE. Output from CNN and viewpoint embedding is concatenated and fed into feedforward network.

We used Planar Flow in this encoder. The dimension of the flow is same as the dimension of $\mathbf{z}^g$, and the transforms are applied 16 times in this model.

## A.2  SEQUENTIAL ENCODER

| Type | input dim | hidden dim (output dim) | Activation | Comment |
|------|-----------|-------------------------|------------|---------|
| LSTM | $\mathbf{z} \times 2 + \mathbf{z}^g$ | 256 | | |
| Linear | 256 | $\mathbf{z} \times 2$ | None | parameters of $\mathbf{z}$ |

Table 4: Architecture of Sequential Encoder.

Sequential Encoder is composed of LSTM and linear layer. Input of LSTM is concatenation of $\mathbf{z}^g$ and $\mathbf{z}$'s parameters inferred in last iteration. Initial input is concatenation of $\mathbf{z}^g$ and zero vector.

## A.3  REFINEMENT NETWORK

| Type | Size | Activation | Comment |
|------|------|------------|---------|
| Input | $\mathbf{z} \times 4 + \mathbf{z}^g \times 2$ | | |
| Linear | 256 | ELU | |
| Linear | 256 | None | |
| LSTM | 32 ($\mathbf{z} \times 2$) | - | |

Table 5: Architecture of Sequential Encoder.

Refinement Network takes log-likelihood, gradient of the parameters (mean and variance) of $\mathbf{z}^{i-1}$ and the parameters of $\mathbf{z}^g$ as input, where $i$ means iteration step.

### A.4 STRUCTURED PRIOR

| Type | size (model dim) | Activation | Comment |
|------|------------------|------------|---------|
| Linear | 256 | ELU | + positional encoding |
| Transformer | 256 | GeLU | Feed forward: 1024
Layers: 3
head: 8 |
| Linear | $\mathbf{z} \times \mathbf{2}$ | None | reparameterization |

Table 6: Architecture of Structured Prior.

Structured prior is implemented by Transformer. We used learnable positinal embedding, and activation is GeLU (Hendrycks & Gimpel, 2016) as in BERT (Devlin et al., 2019). We did not use dropout here.

### A.5 BROADCAST DECODER

| Type | Size/Channels | Activation | Comment |
|------|---------------|------------|---------|
| Input | $\mathbf{z} + \mathbf{v}$ | | |
| Linear | 512 | ReLU | |
| Linear | $\mathbf{z}$ | - | viewpoint projector |
| Broadcast | $\mathbf{z} + 2$ | | |
| Conv $3 \times 3$ | 32 | ReLU | |
| Conv $3 \times 3$ | 32 | ReLU | |
| Conv $3 \times 3$ | 32 | ReLU | |
| Conv $3 \times 3$ | 32 | ReLU | |
| Conv $1 \times 1$ | 4 | ReLU | RGB ch. + mask logit |

Table 7: Architecture of Broadcast Decoder.

We used same decoder as MulMON. $\mathbf{v}$ and $\mathbf{z}$ are concatenated. Concatenated vector is input to projector (former part of the table), then the output is fed into broadcast decoder (latter part of the table).

# B ADDITIONAL RESULTS

## B.1 QUANTITATIVE EVALUATION ON OBSERVATION

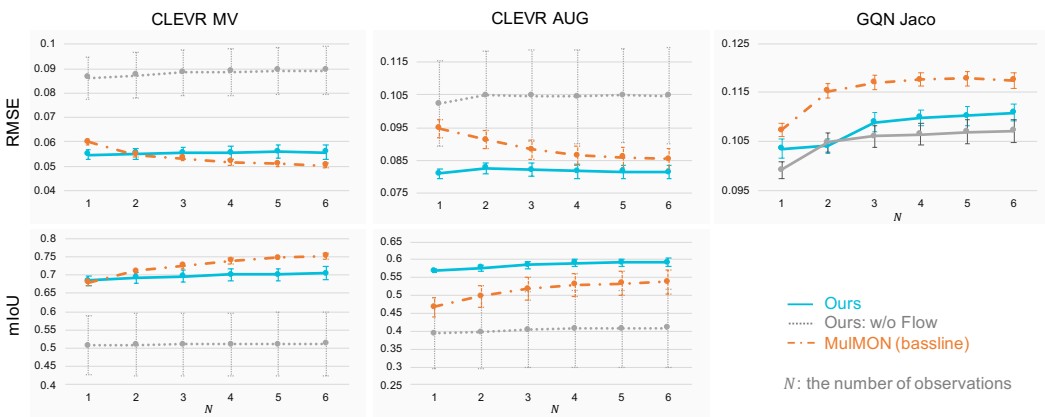

Figure 7: Quantitative evaluation of novel view synthesis. This figure shows scores about observation views, namely, reconstruction and segmentation.

Figure 2 in the main text showed quantitative evaluation for query views. In this section, we show the results for observation views (Figure 7). Namely, this figure is about reconstruction and segmentation quality. The performance of MulMON with small $N$ is relatively not good. This is because MulMON needs to process observation views iteratively, thus the model requires at least several observation views to achieve its best performance. On the other hand, WeLIS is stable against $N$ thanks to the parallel processing of Global Encoder.

## B.2 NOVEL VIEW SYNTHESIS AND SEGMENTATION

### B.2.1 EVALUATION METRICS AND QUALITATIVE DIFFERENCE

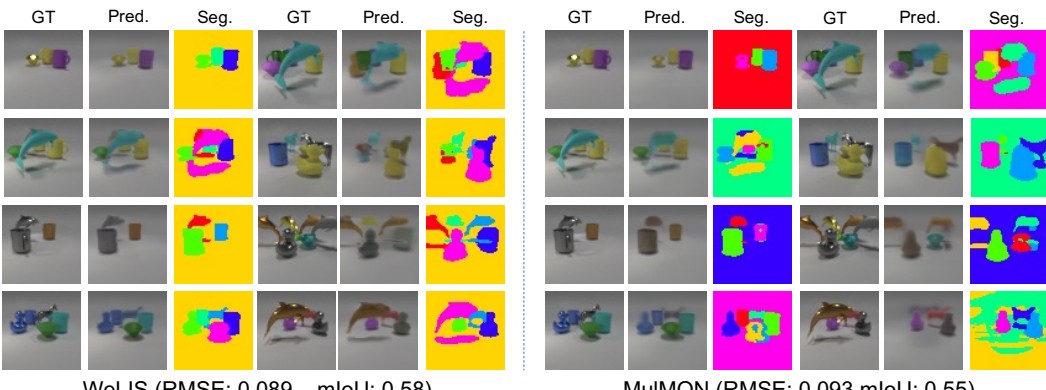

Figure 8: Qualitative results of novel view synthesis. The number of observation views $N$ is three in this figure. The left most columns in each group show ground truth image that is not shown to the model. The second columns in each group show predicted image. The third columns show segmentation mask.

Figure 8 shows additional quantitative samples of novel view synthesis. This figure helps us to understand how much quantitative difference of RMSE and mIoU actually correspond to the qualitative difference.

In addition, we can see that WeLIS represents background with a fixed slot (yellow), while MulMON represents it with different slots (various colors). This is because the auto-regressive encoder is no longer permutation invariant. It can be beneficial that fixed slot represents background, however, disentanglement might be sacrificed with this. It is difficult to conclude which one is better currently, because this kind of properties should be evaluated by various downstream tasks.

## B.3 TRAVERSAL OF GLOBAL LATENT SPACE

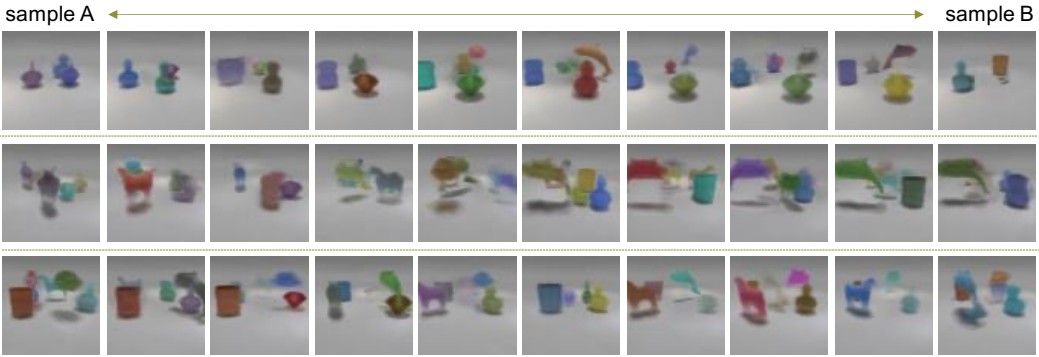

Figure 9: Traversal of global latent space

Figure 9 shows visualization of traversal of global latent space $z^g$. We sample two $z^g$ and visualized interpolation of them. The figure shows three different examples. We can see that spatial arrangement is preserved, but individual components changes quickly.

## B.4 ABLATION STUDY

### B.4.1 NOVEL VIEW SYNTHESIS

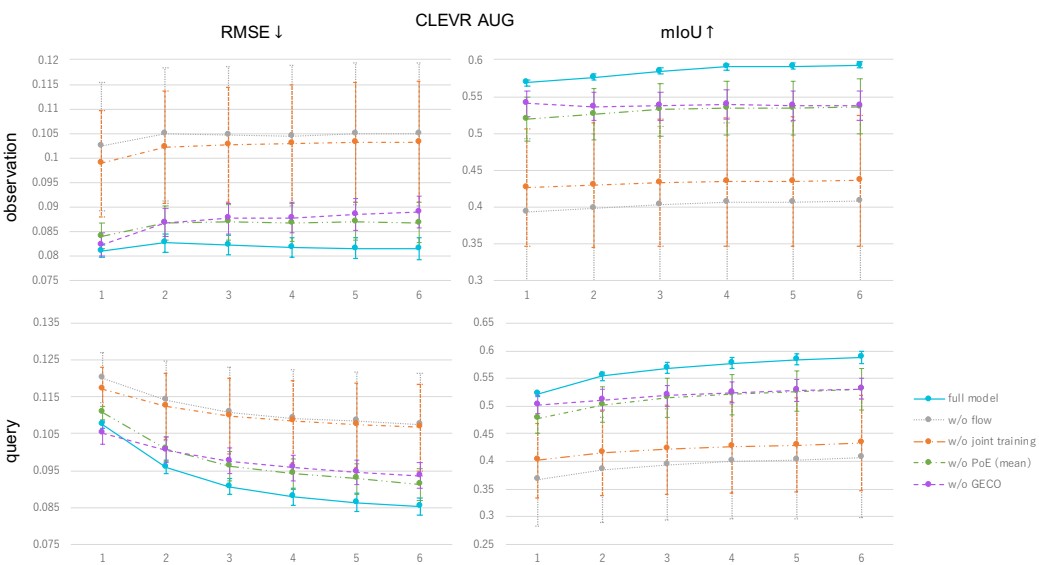

Figure 10: Ablation study for each component: Global Encoder (PoE), GECO, separate training of Structured Prior

Figure 10 shows ablation result of Global Encoder (PoE), GECO, separate training of Structured Prior, and Normalizing Flow. Regarding PoE, we replaced PoE with simple average, which is often used in many researches such as NeRF-VAE (Kosiorek et al., 2021).

This result indicates that each component contributes to the scores and training stability.

### B.4.2 NOVEL SCENE GENERATION

We tested two different architectures for Structured Prior. Both variants use MLP, and one of them updates each slot independently, while the other one considers interaction between slots. As we show in Table 8, there is no significant difference between these variants. This result indicates that $z^g$ alone properly models object placement, and Structured Prior does not have to model dependency between slots.

|     | Transformer | MLP (w/o interaction) | MLP (w/ interaction) |
|-----|-------------|-----------------------|----------------------|
| FID | $83.3 \pm 1.7$ | $84.35 \pm 2.33$ | $84.18 \pm 2.61$ |

Table 8: FID score from two different architectures of Structured Prior

### B.5 DOWNSTREAM TASK

| Models | Datasets | Observations | Accuracy |
|--------|----------|--------------|----------|
| MulMON | | | 64.6 |
| WeLIS($\mathbf{z}^g$) | CLEVR MV | 3 | **73.0** |
| WeLIS($\mathbf{z}$) | | | 63.5 |
| WeLIS($\mathbf{z}^g + \mathbf{z}$) | | | 71.5 |
| MulMON | | | 75.0 |
| WeLIS($\mathbf{z}^g$) | CLEVR MV | 4 | **76.5** |
| WeLIS($\mathbf{z}$) | | | 53.5 |
| WeLIS($\mathbf{z}^g + \mathbf{z}$) | | | 76 |
| MulMON | | | **77.5** |
| WeLIS($\mathbf{z}^g$) | CLEVR MV | 5 | 76.0 |
| WeLIS($\mathbf{z}$) | | | 58.0 |
| WeLIS($\mathbf{z}^g + \mathbf{z}$) | | | 76.0 |
| MulMON | | | 65.4 |
| WeLIS($\mathbf{z}^g$) | CLEVR AUG | 3 | 63.9 |
| WeLIS($\mathbf{z}$) | | | 74.5 |
| WeLIS($\mathbf{z}^g + \mathbf{z}$) | | | **77.5** |
| MulMON | | | 70.7 |
| WeLIS($\mathbf{z}^g$) | CLEVR AUG | 4 | 68.8 |
| WeLIS($\mathbf{z}$) | | | **78.7** |
| WeLIS($\mathbf{z}^g + \mathbf{z}$) | | | 78.3 |
| MulMON | | | 71.8 |
| WeLIS($\mathbf{z}^g$) | CLEVR AUG | 5 | 69.3 |
| WeLIS($\mathbf{z}$) | | | 79.3 |
| WeLIS($\mathbf{z}^g + \mathbf{z}$) | | | **79.6** |

Table 9: Accuracy of a downstream task.Observations stands for the number of available observation views.

We evaluated the latent space of $\mathbf{z}^g$ qualitatively in Figure 6. In this section, we evaluate the obtained representation by downstream task. The task is to predict how many objects are in the observed scenes. We used linear classifier in all the models.

WeLIS can use both $\mathbf{z}$ and $\mathbf{z}^g$ for classification task, thus we tested all three patterns: $\mathbf{z}$ only, $\mathbf{z}^g$ only and both $\mathbf{z}$ and $\mathbf{z}^g$.

## C    DATASETS

CLEVR MV                      CLEVR AUG                      GQN JACO

Figure 11: Ground truth samples of the three datasets used in this paper: CLEVR MV, CLEVR AUG and GQN Jaco.

We show ground truth samples of the three datasets: CLEVR MV, CLEVR AUG and GQN Jaco. These datasets are available from the official implementation of MulMON (`https://github. com/NanboLi/MulMON`).

