# OpenReview forum: "Learning Global Spatial Information for Multi-View Object-Centric Models"
_ICLR.cc/2022/Conference — ICLR 2022 Submitted_

### Official Review · Reviewer_TFws · 2021-10-25

**Correctness:** 3
**Technical Novelty And Significance:** 2
**Empirical Novelty And Significance:** 2
**Recommendation:** 5
**Confidence:** 4

**Main Review:**

Strengths:
* ***Competitive compared to MulMON on the tested datasets***
The model builds upon MulMON, and the empirical results shown in the paper support that the proposed model is competitive and sometimes outpeforms MulMON.

Weaknesses:
* ***Limited comparison to alternative approaches***
The model is mostly compared against MulMON. However there are other models for inferring object-centric scene representations. The authors try to distance their model from approaches that operate using single views such as UORF or ObjSURF on the basis that this model uses multiple views to infer scene representations. While this is technically true, the novel views generated from the proposed approach look quite worse than those of UORF or ObjSURF, while the proposed approach is using more data and thus solving an easier task. Given this, I would encourage the authors to perform a comparison to one of these models on one of the datasets analyzed in those papers and comparing reconstruction quality, to have a sense of whether using multiple views is useful.

* ***Missing a derivation of the training objective***
The authors propose a training objective derived from a lower bound on the likelihood (ELBO). However, the authors do not derive this ELBO, and I believe it is not technically correct as explained in the paper. Since the authors propose a hierarchical latent variable model, certain latent variables (in this case, the object variables $\mathbf{z}$) are conditional and not independent on the global/top latent variable ($\mathbf{z^g}$). When computing the lower bound, we have terms such as $q(\mathbf{z}|\mathbf{z^g})$ which define a gaussian function conditional on $\mathbf{z^g}$. I believe the denominator cannot be factorized into two KL divergences as shown in the paper due to this dependency, and instead we have a double integral on $\mathbf{z}$ and $\mathbf{z^g}$. While in practice this might have small implications since we approximate integrals/KLs with MC sampling with usually a single sample, I would encourage the authors to provide a full derivation of the ELBO that properly justifies the training objective for their hierarchical graphical model.

* ***Key components of the approach are not well-supported, neither theoretically or empirically***
The model is complex, featuring normalizing flows, structured priors and an optimization algorithm that updates different parts of the model in alternance.
The authors do not provide a theoretical justification for these design decision. Futhermore, they also do not provide a proper empirical justification. In particular, while there are some visualizations of model ablations, the results are pretty inconclusive. It is unclear whether the use of normalizing flows provides a significant improvement, and most of the design choices are left without a proper analysis. This could be improved if the authors provided theoretical justifications as to why certain design decisions are needed or are better than more straightforward choices, or with clear empirical ablations that quantitatively show the difference between using or not using components of the model.

**Summary Of The Paper:**

The paper proposes a model for inferring object-centric scene representations from multiple views that explicitly factorizes the representation into global scene properties and objects.
The model builds upon previous object-centric VAEs and is capable of generating novel views of a scene.
The main contribution of the paper is the global-local factorization of the scene representation together with the architectural changes and optimization tricks required to train a VAE with the proposed latent structure.

**Summary Of The Review:**

The proposed approach, while competitive with a previous baseline, has limited theoretical novelty (hirerachical VAEs have been proposed before), and it is unclear what would be the advantages of using the proposed approach compared to alternative approaches. Therefore, I believe the paper is below the acceptance threshold in its current form.

---

> ### Author Response · Authors · 2021-11-22
> **Reply to reviewer TFws from the authors**
>
> We thank the reviewer for the helpful feedbacks. We address each feedback below.
>
> > Limited comparison to alternative approaches
>
> First of all, we had a little misunderstanding about these methods, and we have changed the expression of the section 2. We think the largest difference between MulMON / WeLIS and uORF / ObSuRF is whether they are generative models, and latter methods can not generate novel scenes inherently and they can not obtain low dimensional representation space like VAE based methods. Therefore, of course we will get some insight from comparing WeLIS and these methods, however, we don’t think it is particularly essential for this paper.
>
> > Missing a derivation of the training objective
>
> There was an error about the sign of KL terms in eq (7). It should have been minus instead of plus. Furthermore, the original expression was misleading because expectation of second KL term was omitted. We’ve revised these points.
>
> > Key components of the approach are not well-supported, neither theoretically or empirically
>
> We have added ablation studies about PoE, GECO and separate training of Structured Prior in appendix. The result indicates that introducing each components lead to better scores and stable training.

---

> > ### Comment · Reviewer_TFws · 2021-11-28
> > **Reply to authors**
> >
> > Thank you for the answer. I checked the changes made and I read the rest of the reviews. My concern about the ELBO objective has not been addressed as there is still no derivation for the objective, and in general the paper still feels not well supported. While the paper has improved with the additions, at this point I still find it below the acceptance threshold in agreement with the rest of the reviewers and will keep my current rating.

---

### Official Review · Reviewer_Pvdr · 2021-10-26

**Correctness:** 3
**Technical Novelty And Significance:** 2
**Empirical Novelty And Significance:** 3
**Recommendation:** 5
**Confidence:** 5

**Main Review:**

## Strengths:

The model is a novel extension of MulMON, albeit using fairly standard components and techniques (autoregressive prior; normalising flow posterior). It performs better than MulMON on two tasks -- novel view synthesis (NVS) and a priori scene generation -- both quantitatively and qualitatively.

The method is described clearly. Components and the overall approach are motivated reasonably, and described precisely. There seems to be enough information to reimplement the method, and code is also provided.

## Concerns:

Adding a global scene prior to MulMON seems a rather small contribution, and the 'auxiliary' modifications (e.g. normalising flow posterior) are of minimal novelty. In particular, global scene priors are well-established in the literature (e.g. GENESIS, GNSM, O3V) and clearly necessary for generative modelling of full scenes, so it's not really surprising that adding one to MulMON is beneficial. The only significant difference of this paper is that it operates in the (easier) multi-view setting.

Moreover, the paper indicates that the global prior is useful for NVS, yet it is not trained jointly with the rest of the model, but post-hoc -- hence has no value in guiding the encoder for NVS. This is in contrast to GNSM and O3V, which both train the scene prior jointly with the full model.

NVS should be evaluated quantitatively and qualitatively vs. GQN and ROOTS and/or other SOTA single-view NVS methods, in the (more challenging) single-view case. It should be evaluated vs. NeRF and Object-Compositional NeRF [Yang, ICCV 2021] and/or other SOTA multi-view methods in the multi-view case.

Generation should be evaluated vs. GENESIS and/or other SOTA object-centric generation methods, as well as an appropriate baseline that is not object-centric (e.g. NVAE).

p7: fig 4 -- why do you show MulMON as a generation baseline, when Tab.1 notes that it is not suitable for generation? This gives a false impression of the benefits of the proposed method. This figure should instead show samples from a SOTA model that is designed for generation, e.g. GENESIS -- even if that model is trained in the more-challenging single-view setting.

p8: fig 5 / sec. 4.2.1 -- why does removing the structured prior make the images for WeLIS more blurry? The correct comparison here would be to retrain the model under the assumption of an iid gaussian (or similar) prior for each object and compare results with the structured prior; in that case the individual objects should still look reasonable. And, the purpose of the structured prior is claimed to be to capture global properties of the scene; the statement that it only helps with local appearance, and that the ablated model still places objects correctly, seems to contradict this.

Qualitative results are not very impressive, even on simple datasets of rendered shapes. The method uses a spatial-broadcast decoder, which is known to encourage color segmentation, making it particularly tuned to these easy datasets (with single-colored objects). The paper would be significantly strengthened by evaluation on a more challenging dataset.

p7: tab. 2 -- the two-std intervals for not-bolded values overlap those for bolded values in most cases; please modify the bolding to indicate which values are better up to statistical significance, as currently it is misleading.

## Minor issues / suggestions:

p3: "...generate novel scenes outside of the training distribution..." -- no, outside the finite training set, but inside the training distribution

p4: "...mostly three" -- when is/isn't it three? How is the viewpoint parameterised -- is it a vector in some absolute world-space, or is it relative to one of the camera views?

p4: eq. 3 -- explain here why you introduce a variational distribution on the object latents given z_g (and not depending on the image except via z_g), when the prior model already defines the exact distribution of z|z_g. I assume this is due to the fact (sec. 3.3) that the 'prior' is trained post-hoc (and therefore unable to guide the training of the encoder).

p6: "GECO is not necessary for our model ... but improves generation quality" -- by how much? This should be covered in the ablation study.

p6: sec. 3.3 -- as this describes part of the generative process (not just the inference training, except the last four lines), and is supposed to be a significant contribution of the paper, I think this section should be moved before the discussion of the variational distribution used for training the 'lower' level of the model.

p7: the paragraph introducing the datasets should cite MulMON, as the supplementary states that is where CLEVR-MV and CLEVR-AUG were introduced

There is no discussion of computational cost (training / inference time).

The following should be cited and/or compared against:
- GIRAFFE [Niemeyer, CVPR 2021] -- focuses on generation, but models multi-object scenes, with semi-explicit 3D
- BlockGAN [Nguyen-Phuoc, NeurIPS 2020] -- same as previous
- O3V [Henderson & Lampert, NeurIPS 2020] -- uses multi-view (video) input, and is object-centric, and has a global scene prior
- Structured Generative Modeling of Images with Object Depths and Locations [Anciukevicius et al, ICML OOL workshop 2020] -- also hierarchical, object-centric, and stage-wise trained (albeit in more challenging single-view setting)

A number of references cite arXiv only, even though the papers are published. Please add the proper conference citations.

**Summary Of The Paper:**

The paper proposes a new structured generative model of multi-view images, representing their latent decomposition into objects. This model extents MulMON -- a spatial mixture model over multi-view images that is suitable for image decomposition but lacks a prior necessary for synthesis -- by adding such a prior (using an autoregressive model), and making some other architectural changes (e.g. adding a normalising flow posterior instead of gaussian). In the proposed model, a global latent variable conditions the sampling of per-object latents, which in turn parameterise a spatial mixture model for each image. The model is evaluated on novel view synthesis (NVS) and a priori scene synthesis, on simple datasets of renderings (e.g. multi-viewpoint CLEVR); it is shown to out-perform a baseline (MulMON) on several metrics for both tasks quantitatively, and qualitative results are also displayed.

**Summary Of The Review:**

The proposed method is a straightforward but original combination of existing techniques. It improves over MulMON on NVS and generation, but the evaluation needs to be much more comprehensive, in particular adding comparisons vs. appropriate baselines for each task. Moreover, certain important technical points regarding the benefit of the model structure should be clarified.

---

> ### Author Response · Authors · 2021-11-22
> **Reply to reviewer Pvdr from the authors 1**
>
> We thank the reviewer for through feedbacks and useful suggestions.
>
> Before addressing each feedback, we need to emphasize that our proposed model is for object-centric NVS, and also for sampling novel scenes.
> Not only unsupervised segmentation, but also (object-centric) NVS and obtaining 3d-aware object representation is also important task which has many potential applications such as robotics.
> Using multi-view input is not just for making segmentation easier, but for NVS and representation learning.
>
> > "The only significant difference of this paper is that it operates in the (easier) multi-view setting.” “(more challenging) single-view case."
>
> Thus, we think focusing only on unsupervised segmentation and claiming MOMV setting as “easier task" is not reasonable criticism.
>
> ---
>
> > In particular, global scene priors are well-established in the literature (e.g. GENESIS, GNSM, O3V) and clearly necessary for generative modelling of full scenes, so it's not really surprising that adding one to MulMON is beneficial.
>
> We agree with the point that introducing global latent is important for generative modeling but not novel, and we did not argue this point as large contribution. The important thing is that we adapt this to multi-view object-centric VAE with several nontrivial techniques.
>
> > The only significant difference of this paper is that it operates in the (easier) multi-view setting.
> Moreover, the paper indicates that the global prior is useful for NVS, yet it is not trained jointly with the rest of the model, but post-hoc -- hence has no value in guiding the encoder for NVS. This is in contrast to GNSM and O3V, which both train the scene prior jointly with the full model.
>
> First of all, the prior which is not trained jointly is the Structured Prior, not the scene prior p(z^g).
> As for O3V, it only has single latent variable for whole scene like basic VAE and GQN, and it does not have hierarchical prior. Thus, the prior of O3V does not guide the encoder, but the encoder itself has hierarchical nature, and it is considered to be important for the model. By the way, O3V only takes fixed number of frames, or viewpoints  as input, thus the problem setting is slightly different from MOMV.
>
> On the other hand, GNM has hierarchical prior and it is trained jointly. The difference between WeLIS and GNM is the inference model. GNM uses mean-field approximation to factorize the generative model into two (non-hierarchical) encoders, but WeLIS has both hierarchical encoder: Sequential Encoder and hierarchical prior: Structured Prior. We added ablation study of separate training in appendix, and we confirmed that the training become slightly unstable in that case.
>
> In conclusion, there is no support from hierarchical prior in O3V and WeLIS, but they make use of their hierarchical encoder. Also, we empirically confirmed that separate training is not harmful (and even helpful) to NVS except for novel scene generation.
>
> > NVS should be evaluated quantitatively and qualitatively vs. GQN and ROOTS and/or other SOTA single-view NVS methods, in the (more challenging) single-view case. It should be evaluated vs. NeRF and Object-Compositional NeRF [Yang, ICCV 2021] and/or other SOTA multi-view methods in the multi-view case.
> Generation should be evaluated vs. GENESIS and/or other SOTA object-centric generation methods, as well as an appropriate baseline that is not object-centric (e.g. NVAE).
>
> The comparison of MulMON to GQN / MulMON to IODINE has already done in MulMON paper, and MulMON performed a lot better than them. Comparison is of course helpful in some aspect, but we don’t think it’s essential to compare models that have different functionality in this paper. As for NeRF based methods you mentioned, both model can not obtain low dimensional representation space and also can not generate novel samples. Especially for Object-Compositional NeRF, the model uses 2D segmentation mask as input, thus the objective of the methods is totally different in our understanding.

---

> ### Author Response · Authors · 2021-11-22
> **Reply to reviewer Pvdr from the authors 2**
>
> > p7: fig 4 -- why do you show MulMON as a generation baseline...
>
> MulMON and our model is designed for object-centric NVS and obtaining object representations, and our model is also for (object-centric view-conditional) novel scene generation in addition to that. If we ignore NVS and focusing only on unsupervised segmentation, it can be “mode-challenging single-view setting”, but it is not the case. Moreover, one of the objective of WeLIS is to extend MulMON for novel scene generation, thus we are sure that this comparison is not “false impression of the benefits” at all. However, at the same time, we agree with the point that we don’t need to emphasize the generation results from MulMON so much, thus we are planning to change the size or amount of the content of the figure.
>
> > p8: fig 5 / sec. 4.2.1 -- why does removing the structured prior make the images for WeLIS more blurry? The correct comparison here would be to retrain the model under the assumption of an iid gaussian (or similar) prior for each object and compare results with the structured prior...
>
> As we mentioned in the paper, IAI bring about this. The updated posteriors of each object can not be sampled, because of the iterative update. Moreover, even though MulMON applies iid Gaussian to updated posteriors as penalty term, sampled objects are still blurry. Therefore we address this problem by introducing Structured Prior which models updated posterior.
>
> >  And, the purpose of the structured prior is claimed to be to capture global properties of the scene; the statement that it only helps with local appearance, and that the ablated model still places objects correctly, seems to contradict this.
>
> We tested two types of Structured Prior which are implemented as MLP. One of them does not have interaction between slots, and the other does like transformer. As a result, there are no significant difference between them. This indicates that Structured Prior does not have to model interaction between slots, and thus we’ve changed the expression about that point as you pointed out. This result is added to appendix.
>
> MLP (iid): 84.35 $\pm$ 2.33
> MLP (interaction between slots): 84.18 $\pm$ 2.61
>
> > Qualitative results are not very impressive, even on simple datasets of rendered shapes...
>
> We know that broadcast decoder does not scale to realistic images currently. We think current options for a decoder is broadcast decoder, neurosymbolic decoder originates from  Attend, Infer, Repeat and recent NIR based methods. However, as far as we know, there are no generative models which use NIR and can perform both object-centric inference and sampling. Therefore we are sure that doing researches using broadcast decoder is not useless even if it does not scale to real images currently.
>
> > p7: tab. 2 -- the two-std intervals for not-bolded values overlap those for bolded values in most cases; please modify the bolding to indicate which values are better up to statistical significance, as currently it is misleading.
>
> We have changed the notation.
>
>
> ---
>
> minor issues
>
> ---
>
> We have changed the expressions and citations according to your suggestion.
>
> > the dimension of viewpoint vectors
>
> A viewpoint of CLEVR-MV and CLEVR-AUG always faces the center of the scenes, and they are represented as polar coordinates. Thus, it is V = (cos \alpha, sin \alpha, r), where \alpha is the azimuth angle and r is the distance to the center of the scenes.
>
> > p4: eq. 3 -- explain here why you introduce a variational distribution on the object latents given z_g (and not depending on the image except via z_g), when the prior model already defines the exact distribution of z|z_g. I assume this is due to the fact (sec. 3.3) that the 'prior' is trained post-hoc (and therefore unable to guide the training of the encoder).
>
> In GENESIS and GNM, one of their latent variables is explicitly bound to segmentation masks or neurosymbolic variables such as parameters of affine transform. In this case, we can model two latent variables independently, and can use two different encoders. However, two latent variables in WeLIS are not bound to particular role, and thus just using two encoders can bring about posterior collapse. This is why we used hierarchical encoder. However we agree with the point that optimal modeling should be explored more intensively.
>
> > p6: "GECO is not necessary for our model ... but improves generation quality" -- by how much? This should be covered in the ablation study.
>
> We’ve also added an ablation study of GECO in appendix.

---

> > ### Comment · Reviewer_Pvdr · 2021-11-29
> > **Reply to authors**
> >
> > Thanks authors for the detailed responses to my review, and for the updated manuscript. These address some of my concerns, yet others remain (as do some from the other reviewers). Hence, I upgrade my rating slightly, but still feel that the paper would benefit from additional work before publication. In particular, as mentioned before, the evaluation is too much focused on comparison with MulMON, without considering other state-of-the-art methods for the tasks considered. As an additional suggestion, I think it would be clearer if you add separate figures showing the graphical model for the generative and for the variational -- this would maybe resolve some concerns about how exactly the various components interact. The exact relation between the 'structured prior' and the 'sequential encoder' remains unclear to me even after reading several times.

---

### Official Review · Reviewer_RJ1B · 2021-10-31

**Correctness:** 3
**Technical Novelty And Significance:** 2
**Empirical Novelty And Significance:** 2
**Recommendation:** 5
**Confidence:** 3

**Main Review:**

# Strengths
- The paper applies normalizing flows to stable inference and training, which might be empirically useful.

# Weaknesses/Questions
1. Introducing a global latent variable does not necessarily model spatial information. However, the authors seem to claim a straightforward global latent variable without any explicit modeling can tackle this issue. Even if it might be empirically true, it is better to change the tone, e.g. "we introduce a global latent variable and empirically find that it can model spatial information, which is shown in Sec xxx". Otherwise, it is misleading to readers. Or the authors can claim that previous methods may ignore modeling global relationships while this paper takes it into consideration.
Similarly, the authors also claim their model can generate physically plausible novel scenes. It seems that the authors mainly rely on the Structured Prior (Transformer) to achieve this, but there is no reasonable analysis about why the Structured Prior is relevant to physical plausibility.

2. The equation (7) looks erroneous. The first KL term seems to be derived from $E_{q(z,z^g|X)} [\log \frac{p(z^g)}{q_\phi (z^g|X)}]$. However, $KL[q_\phi (z^g|X)||p(z^g)] = E_{q(z^g|X)} [\log \frac{q_\phi (z^g|X)}{p(z^g)}]$. Can the authors check the correctness of the ELBO used in the paper?

3. The baselines of single-view object-centric learning methods, e.g. IODINE, MONET, GENESIS, are missing. Although the setting is multi-view, those baselines are still comparable.

**Summary Of The Paper:**

The paper tackles object-centric learning from multiple views. It proposes an extension to MulMON. Concretely, it introduces a global latent variable, aiming at modeling global information like spatial relationships, and predicts object-level latent variables upon the global one by iterative amortized inference. It also uses normalizing flows to stabilize training and inference. The authors compare the proposed model with  MulMON on three simple datasets: CLEVR MV, CLEVR AUG, GQN Jaco.

**Summary Of The Review:**

The paper does not fully justify why the introduced hierarchical probabilistic model can lead to their claims: spatial information and physical plausibility, which makes those claims subjective. Besides, the baselines are limited and thus experiments are not convincing. Overall, I tend to reject this paper.

---

> ### Author Response · Authors · 2021-11-22
> **Reply to reviewer RJ1B from the authors**
>
> We thank the reviewer for the helpful feedbacks. \
> First of all, your summary and comments did not mention novel view synthesis(NVS), but one of the important task of MulMON and WeLIS is object-centric NVS, not only unsupervised segmentation. Thus we believe our model has more strength and novelty other than using normalizing flow for stable training.
>
> > Introducing a global latent variable does not necessarily model spatial information.
>
> We agree with this point. However, a latent variable which can represent spatial information of the whole scene should summarize or integrate object-level representations. Thus we consider that introducing global latent variable can be a reasonable solution for this, and we empirically confirmed this.
>
> > The equation (7) looks erroneous.
>
> As you pointed out, equation (7) had an error about sign, and some part was unclear.
> Firstly, we corrected the sign of KL terms. It should have been minus. Secondly, missing expectation in the second KL term led ambiguity. We also revised this point.
>
> > The baselines of single-view object-centric learning methods...
>
> Comparison against IODINE and GQN has been done in MulMON already, thus we did not do that in this paper. Thanks to the multi-view setting, MulMON has significantly  better results than IODINE and GQN in terms of both segmentation and NVS.

---

> > ### Comment · Reviewer_RJ1B · 2021-11-29
> > **Reply to authors**
> >
> > Thanks for the authors' responses. I would like to slightly improve my score. However, the correctness of the ELBO is still not verified through a step-by-step derivation, which is also mentioned by another reviewer (TFws). Besides, the contribution of the paper is still limited, or at least ambiguous.

---

### Official Review · Reviewer_KiHp · 2021-11-02

**Correctness:** 3
**Technical Novelty And Significance:** 2
**Empirical Novelty And Significance:** 2
**Recommendation:** 5
**Confidence:** 4

**Main Review:**

Strengths:

* the global variable seems to help capture global scene structure
* decent analysis with ablations and latent investigation
* acceptable results for reconstruction but at the same time nice samples

Weaknesses:

* One thing that is not clear is the interaction between GECO and IAI - it seems to me that this is not trivial to achieve as the lagrange multipliers would need to be maintained both internally and externally in the losses. Can the authors elaborate on that point?
* What is the purpose of the PoE encoder over the views? it's not immediately clear to me why would that help - there are other ways to handle multiple views which may work as well (see for example NeRF-VAE - Kosoirek et al.).
* I would be happy to see latent traversals for the global latents - while we can see they capture structure of scenes it would be interesting to see how they change and how the resulting scene changes.
* What happens when you instantiate the model with different number of slots? does the global latent overfit to the number or does it adapt? what happens when you test on scenes with different number of objects?
* I feel experimental validation is quite limited - both in dataset use (would be nice to see more complex data with more complex structure globally) and evaluation metrics.


**Summary Of The Paper:**

This paper presents a hierarchical, object centric view conditional generative model for scenes. The latent structure includes a global latent variable which captures general scene structure/configuration and conditions a set of slotted latents, each one capturing a single object in the scene. An inference method is proposed - a product of expert + flow encoder for the global, and a sequential encoder + iterative refinement for the slots. Everything is view conditional so training requires camera information, but novel views may be synthesized at ease.
The model is demonstrated to work on CLEVR and a slightly more complex version of CLEVR with more object structures as well as the Jaco arm dataset.

**Summary Of The Review:**

All in all this is a nice paper, with a bit more depth could be a nice contribution.

---

> ### Author Response · Authors · 2021-11-22
> **Reply to reviewer KiHp from the authors**
>
> We thank the reviewer for the constructive feedback and suggestions. We address the questions and suggestions below.
>
> > One thing that is not clear is the interaction between GECO and IAI - it seems to me that this is not trivial to achieve as the lagrange multipliers would need to be maintained both internally and externally in the losses. Can the authors elaborate on that point?
>
> GECO and IAI does not conflict. IAI updates NLL based on its gradient and several auxiliary inputs, and the updated NLL is passed to GECO. Therefore, GECO does not have interaction with the IAI's process.
>
> > What is the purpose of the PoE encoder over the views? it's not immediately clear to me why would that help - there are other ways to handle multiple views which may work as well (see for example NeRF-VAE - Kosoirek et al.).
>
> Most of the methods like GQN and NeRF-VAE uses mean or sum to aggregate multiple views, and as you pointed out,  this method can also be applied to our model. However, this simple aggregation method is not sufficient for our global latent variable which is expected to be expressive. In the context of multi-modal VAE, PoE and Mixture of Experts (MoE) are often used to aggregate multi-modal information, and our model is inspired by these methods (MVAE [Wu 2019], MMVAE[Shi 2019]). In this research, we chose PoE because the resulting distribution will not be sharper than individual experts in MoE. This is considered to be not desirable for multi-view aggregation.
>
> To address this point, we added ablation study in appendix. We compare PoE and simple mean, and confirmed PoE performs better and the training is stabilized as well.
>
> > I would be happy to see latent traversals for the global latents - while we can see they capture structure of scenes it would be interesting to see how they change and how the resulting scene changes.
>
> The result of latent traversal is added in appendix. Spatial arrangement is preserved with traversal, but individual components changes quickly.
>
> > What happens when you instantiate the model with different number of slots? does the global latent overfit to the number or does it adapt? what happens when you test on scenes with different number of objects?
>
> Though we fixed the number of slots (K) at seven in this paper, there is no problem using different K. However, we did not evaluated generalization of K at test time because of two reasons. First one is that it is not considered to be important for application scenarios. Even if we fix K, excess slots do nothing and do not harm final result, and we just need to set large K enough for the objective. The other one is that we don't have test set which has different number of objects from train set currently.
>
> > I feel experimental validation is quite limited - both in dataset use (would be nice to see more complex data with more complex structure globally) and evaluation metrics.
>
> We already have three datasets which is used in MulMON, and we believe this is enough in terms of a comparison. Thus we did not add new dataset this time.

---

> > ### Comment · Reviewer_KiHp · 2021-11-29
> > **Thank you for the responses**
> >
> > Regarding GECO - I would clarify, in that case, that the loss used internally in IAI and the loss optimised externally are different.
> >
> > Regarding the latent traversals - I must say these are not convincing me when it comes to capturing global structure, I would expect to actually see the individual components to remain more or less fixed and just the global structure to change. Both the number and spatial organisation of object changes considerably there, and not in a particularly smooth way - this could be because of the way the traversal is generated but again, seems to point against the latent learning global structure.
> >
> > Regarding number of slots - I think this is actually an important point, because there is a non-trivial relationship between the global latent and the number of slots. It's not clear to me how the model would adapt to different number of slots and how the resulting global latent would differ. I still think this was worth investigating.

---

### Author Response · Authors · 2021-11-22
**Summary of revisions**

We thank all the reviewers for great feedbacks. We list major changes below as a summary.

* Added ablation studies \
 We have added ablation studies of  PoE, GECO and separate training of Structured Prior in appendix. We think this results clalified the contributions of each component.

* Tested different Structured Prior architecture \
We also tested different architectures of the Structured Prior, and added a result in appendix. We tried two variants and both of them are implemeted as MLP. One of them updates each slot independently while the other one consider interaction between slots. The result indicates that the Structured Prior does not have to model dependency between slots. However, transfomer version performed the best anyway.

* Revised section 2 (related work) \
We have changed the expression about "Slot Attention + NeRF" methods, i.e., uORF and ObSuRF. They can scale to more realistic images and have great quality, but the point is that they are not generative model. Therefore, they can not generate novel scenes inherently, and can not obtain low dimensional representation space or latent space.

* Revised section 3.3 (structured prior) \
According to the result of additional experiment on Structured Prior, we changed the expression about it.

* Revised training objective \
The original version had typo (the signs of KL terms), and expectation was omitted. We revised this point for clality.

* About additional experiments \
Some reviewers suggested adding other datasets, however, we already have three datasets which is used in MulMON, and we believe this is enough for this paper.

---

### Decision · Program_Chairs · 2022-01-20

**Decision:**

Reject

**Comment:**

This submission received four high-quality reviews and there are a lot of meaningful discussions during the author response period. After the discussions, all four reviewers agreed that the submission can be strengthened in a number of ways, including more solid experimental results and a justification for the correctness of the ELBO.  The AC agrees. The authors are encouraged to revise the paper based on the reviews for the next venue.